# An oral pH-responsive *Streptococcus agalactiae* vaccine formulation provides protective immunity to pathogen challenge in tilapia: A proof-of-concept study

Shazia Bashir[1☯], Nguyen Ngoc Phuoc[2☯], Tharangani Herath[3], Abdul Basit[1], Ruth N. Zadoks[4]*, Sudaxshina Murdan[1]*

1 UCL School of Pharmacy, London, United Kingdom, 2 Faculty of Fishery, Hue University of Agriculture and Forestry, Hue University, Hue, Vietnam, 3 Department of Animal Health, Behavior and Welfare, Harper Adams University, Newport, Shropshire, United Kingdom, 4 Faculty of Science, Sydney School of Veterinary Science, The University of Sydney, Camden, NSW, Australia

☯ These authors contributed equally to this work.
* Ruth.zadoks@sydney.edu.au (RNZ); s.murdan@ucl.ac.uk (SM)

**Data Availability Statement:** Data are available at: Murdan, Sudax; Phuoc, Nguyen Ngoc; Basit, Abdul; N. Zadoks, Ruth; Herath, Tharangani;

## Abstract

Intensive tilapia farming has contributed significantly to food security as well as to the emergence of novel pathogens. This includes *Streptococcus agalactiae* or Group B *Streptococcus* (GBS) sequence type (ST) 283, which caused the first known outbreak of foodborne GBS illness in humans. An oral, easy-to-administer fish vaccine is needed to reduce losses in fish production and the risk of zoonotic transmission associated with GBS. We conducted a proof-of-concept study to develop an oral vaccine formulation that would only release its vaccine cargo at the site of action, i.e., in the fish gastrointestinal tract, and to evaluate whether it provided protection from experimental challenge with GBS. Formalin-inactivated *S. agalactiae* ST283, was entrapped within microparticles of Eudragit® E100 polymer using a double-emulsification solvent evaporation method. Exposure to an acidic medium simulating the environment in tilapia stomach showed that the size of the vaccine-loaded microparticles decreased rapidly, reflecting microparticle erosion and release of the vaccine cargo. *In vivo* studies in tilapia showed that oral administration of vaccine-loaded microparticles to fish provided significant protection from subsequent homologous pathogen challenge with GBS ST283 by immersion compared to the control groups which received blank microparticles or buffer, reducing mortality from 70% to 20%. The high efficacy shows the promise of the vaccine platform developed herein, which might be adapted for other bacterial pathogens and other fish species.

## 1. Introduction

Aquaculture is a crucial part of the economy of low and-middle income countries (LMICs), including Vietnam, employing and feeding millions of people [1]. Tilapia (*Oreochromis* spp.)–

Bashir, Shazia (2022): An oral pH-responsive
Streptococcus agalactiae vaccine formulation
provides protective immunity to pathogen
challenge in tilapia. University College London.
Dataset. https://doi.org/10.5522/04/19369376.v1.

**Funding:** Financial support was obtained from
University College London (UCL), via an internal
UCL award funded through Research England's
'QR Global Challenges Research Fund'.
Furthermore, UCL, Hue University (https://hueuni.
edu.vn/portal/en/), The University of Sydney
(https://www.sydney.edu.au/) and Harper Adams
University (https://www.harper-adams.ac.uk/)
funded SM's, NNP's, RZ's and TH's time,
respectively, on the project. We also thank the Hue
University Strong Research Group (NCM.
DHH.2022.05) for staff and facilities support. The
funders had no role in study design, data collection
and analysis, decision to publish, or preparation of
the manuscript. Dr Shazia Bashir received a salary
from University College London.

**Competing interests:** The authors have declared
that no competing interests exist.

is eminently suitable for aquaculture due to its fast growth, ability to survive in poor water conditions, to breed throughout the year, to produce high numbers of eggs, and to eat a wide range of feeds [2], and is the third most common fish species in aquaculture [3] with global production volumes reaching 4.5 million tonnes in 2018. In addition to producing more food, intensification of tilapia farming also increases the risk of infectious disease outbreaks, causing significant waste and economic loss. Streptococcosis, the most important disease of tilapia [4], is primarily caused by *Streptococcus agalactiae*, known in human medicine as group B *Streptococcus* (GBS). Streptococcosis mostly occurs when fish are stressed due to, for example, an increase in water temperature, sub-optimal oxygen levels or overcrowding and can lead to high levels of disease and mortality, especially towards the end of the production cycle, aggravating the losses incurred [4,5]. In 2015, GBS sequence type (ST)283 caused an outbreak of invasive disease in people in Singapore. Uniquely, those affected had few underlying comorbidities, and the route of exposure was foodborne, traced back to the consumption of raw fish with GBS [5]. Based on evolutionary analysis of genomic data, GBS sequence type (ST) 283 emerged around the time of intensification of aquaculture in the early 1980s and is now a recognized cause of severe invasive disease in both fish and humans in Southeast Asia [6,7]. Its unusual–foodborne—route of transmission has led the Food and Agricultural Organisation (FAO) of the United Nations to publish a Risk Profile on its characteristics and role as a human health hazard in 2021 [8].

To control disease in fish, antimicrobials are commonly used, especially in LMICs [1,9]. This is undesirable, both from fish, and public, health perspectives, as sick fish have diminished appetite and therefore do not benefit from oral antimicrobial treatment, whilst the release of antimicrobials in the aquatic environment may exert selection for antimicrobial-resistant pathogens [1,10,11]. Prophylactic vaccination of fish is a far more desirable alternative [12], and many types of fish vaccines have been developed, including vaccines for delivery by injection, immersion, orally, or as spray [13]. Most vaccines that are available for tilapia are injectable formulations [14,15], making them suitable for fish that are large enough to be handled and anaesthetized. This process demands individual handling of fish, which can lead to handling-related stress and immunosuppression, which may have a negative impact on vaccine response, and even cause mortalities [14]. The process is also labour-intensive, making it cost-prohibitive for low value fish such as tilapia. Oral vaccination is possibly a better method for mass immunization of high volume, low value species such as tilapia. It has been proven to be effective in protecting against mass mortality [16]. In addition, oral administration stimulates mucosal immunity, i.e., the appearance of antigen specific antibodies in skin, mucus, bile, or intestine [17,18], which is generally where the first contact between aquatic pathogens and their hosts occurs [19,20].

To enhance the efficacy of oral vaccines, several strategies have been evaluated, many of which focus on inhibiting antigen degradation in the fish stomach. Such methods include the concomitant administration of antacids and antiproteases, use of 'prills' (pellets with lyophilized vaccine incorporated into a matrix of saturated long chain fatty acids), coating of vaccine granules, beads and vaccine-coated feed pellets with acid-resistant polymers, and encapsulation of vaccine antigens into vaccine carriers such as microparticles, beads or liposomes [21–35]. The pH-responsive vaccine carriers are incorporated into food pellets that are fed to the fish during regular feeding times. This approach bypasses the fish stomach to avoid exposure of the antigen to stomach acid and enzymatic degradation because the vaccine carrier will only dissolve in the fish intestines. However, polymers that are soluble in the intestinal environment of the fish (where pH is close to neutral) may also be soluble in water (where pH is also close to neutral) and vaccines fabricated from these types of polymers may leach from the feed into the aquatic environment, resulting in loss of antigen. To overcome such losses and induce

appropriate levels of immune protection, fish must be fed with vaccine-loaded feed for multiple consecutive days. For example, Halimi *et al.* [35] developed vaccine-loaded fish feed pellets coated with Eudragit L30D-55 for rainbow trout. The relative percentage survival (RPS) was found to be very high for the vaccinated groups (85 ± 7% for *Streptococcus iniae*, 72 ± 8% for *Lactococcus garviae*) in comparison to the unvaccinated group, but this was only achieved with daily immunization for 14 consecutive days. Hayat *et al.* [36] assessed the effectiveness of a formalin-killed whole cell *S. iniae* vaccine loaded within food pellets in red hybrid tilapia. Several immunization regimens were tested, and the lowest mortality was achieved for the fish group that were orally vaccinated by feeding for 9 consecutive days and then received oral boosters (by gavage) on days 14 and 21. The need for such prolonged vaccine-feeding detracts from the potential cost-effectiveness of oral vaccines.

To address the limitations of current feed-based vaccine delivery, we investigated a different approach to those outlined above, and developed a prototype vaccine that will dissolve in the fish stomach but not in the aquatic environment, by entrapping the vaccine in a matrix of Eudragit E100. This polymer, was chosen because it is generally recognized as safe (GRAS), enabling its use in food-producing animal species [37]. In this paper, we report on the preparation and characterization of vaccine-loaded Eudragit E100 microparticles and an evaluation of vaccine efficacy using an experimental challenge model. We show that our approach protects tilapia from homologous challenge with zoonotic GBS ST283 and discuss opportunities for further development of this successful prototype.

## 2. Materials and methods

### 2.1 Bacteria

The vaccine strain, GBS isolate MRI Z2-388, belongs to ST283, and was originally isolated by the authors from the brain of a clinically affected tilapia in the Mekong Delta area, Vietnam, in 2016. To preserve the structural integrity of the bacteria, they were formalin-inactivated in 2% paraformaldehyde (1 h at room temperature, then 24 h at 4°C), followed by suspension to a calculated concentration of $10^9$ colony forming units (CFU)/mL in 0.1 M PBS, pH 7.4.

The challenge bacteria, GBS isolate 0101030, also belongs to the ST283 strain, and was also isolated by the authors from the brain of a clinically affected tilapia in the Mekong Delta area, Vietnam. Both isolates were identified to species and strain level by colony morphology, Gram stain, and multi-locus sequence typing (MLST) as described in [6].

To determine the morphological characteristics of the formalin-inactivated bacteria used in the vaccine formulation, samples of a dilute dispersion of GBS (in de-ionized water, $10^7$ CFU/ml) were mounted onto aluminum stubs and allowed to dry overnight before being sputter coated with gold in a high-vacuum evaporator for 3 min at 30mA (Emitech K550, Ashford, England) and photographed using a scanning electron microscope (SEM, Philips XL30, Eindhoven, Holland). The particle size and zeta potential of GBS organisms was measured by dynamic light scattering using the Zeta sizer Ultra (Malvern instruments Ltd, Worcestershire, UK), using a GBS suspension (at a concentration of $10^6$ CFU/mL in deionized water).

### 2.2 Preparation and characterization of microparticles

Eudragit E100 was kindly gifted by Evonik (Essen, Germany). Polyvinyl alcohol (PVA) was purchased from BDH Laboratory supplies (Poole, England), Sorbitan monostearate from Thermo Fisher Scientific (Heysham, UK), and Absolute Ethanol (≥99.8%) and Dichloromethane from Sigma-Aldrich, UK. All materials were of laboratory grade quality.

Both blank and GBS-loaded microparticles were produced using the emulsification solvent evaporation (ESE) technique. Blank microparticles were prepared by slowly adding 3 mL of a

solution of Eudragit E100 (7.6%w/v) in dichloromethane-ethanol (1:1 v/v) dropwise (over 30 min) to an aqueous solution of PVA (6 mL, 4% w/v) at 800 rpm. The mixture was left to stir overnight for solvent evaporation to occur and for microparticles to harden. The resultant microparticles were washed, lyophilized for 48 h at -85°C (Alpha 1–4 LD plus, Christ, Germany), then stored at room temperature until further use.

GBS-loaded microparticles were prepared using a water-in-oil-in-water (w/o/w) double-emulsion solvent evaporation technique adapted from [38]. The primary emulsion was formed by adding 300 μL of GBS suspension ($10^9$ CFU/mL) dropwise (slowly over a 30min period) to an organic solution (3mL) of Eudragit E100 (7.6% w/v) and sorbitan monostearate (2.5% w/v) dissolved in a dichloromethane-ethanol (1:1 v/v) mixture under continuous stirring at 800 rpm. The resultant primary emulsion was homogenized for 2 min (6000rpm, IKA T10 basic Ultra-Turrax), then added dropwise (over a period of approximately 30 min), to an aqueous 4% w/v PVA solution (6mL) under vigorous stirring at 800 rpm. The emulsion was left to stir overnight at the same speed for the solvent to evaporate and the microparticles to harden. The resultant microparticles were washed and rinsed twice (with 0.1 M PBS, pH 7.4) by centrifuging at 10,000 g for 10 min (Sigma Laborzentrifugen GmbH, Germany). After every wash cycle, the supernatant was discarded to remove excess PVA and replaced with fresh 0.1M PBS. The microparticle pellet was collected, lyophilized, and stored, as described for blank microparticles.

To characterize the blank and GBS-loaded microparticles, a mass of lyophilized microparticles (~0.1mg) was added to 10mL of 0.1M PBS and the mixture was manually agitated to obtain a homogenous dispersion that appears sightly opaque. A sample of the resultant suspension was characterized for size and surface charge using the Zeta sizer Ultra (Malvern instruments Ltd, Worcestershire, UK). In addition, microparticle morphology and surface topography was examined by SEM as described for the bacteria.

The pH-responsiveness of Eudragit E100 microparticles was evaluated by placing the particles in an acidic medium (to simulate the conditions of the fasted fish stomach) and monitoring the changes in particle size with time, using the Zeta sizer Ultra. Briefly, 500μL of a microparticle dispersion (2mg in 10mL of de-ionised water) was added to 0.1 M hydrochloric acid (10mL, pH 1.3) under constant stirring at 300rpm. At time intervals, aliquots (0.7mL) were withdrawn from the mixture and the particle size was measured using the Zeta sizer Ultra. Removed aliquots were replaced with an equivalent volume of 0.1 M HCl to maintain sink conditions.

### 2.3 Tilapia studies

Red tilapia (*Oreochromis* sp.) were purchased from the Provincial Breeding Centre of Thua Thien Hue at Cu Chanh Commune, Thua Thien Hue province, Vietnam. The fish were starved for 24 h prior to transportation and maintained at 15°C during the 30-minute journey to the Laboratory of Fish Pathology, Hue University of Agriculture and Forestry (HUAF), Hue City, Vietnam. Fish were housed in 1000 L fiber-glass tanks using continuous flow-through water at a flow rate of 0.38 L min$^{-1}$ at 28°C ± 2°C, and fed with commercial tilapia diet (Aquaxcel 7444, Cargill, Vietnam) at 2% of body weight for 14 days prior to commencing experiments. Mean body weight of the tilapia used in this study was 15 ± 2 g on the day of the vaccination, close to that used in [6]. Before vaccination, the population was tested for *S. agalactiae* based on clinical monitoring and by directly streaking the kidney and the brain of 5 euthanised fish onto tryptone soya agar (TSA, Himedia) as previously described in [6]. All experiments were conducted with approval of The Animal Ethics Committee of Hue University, and in accordance with UK Home Office standards. Written consent was also obtained from University College London.

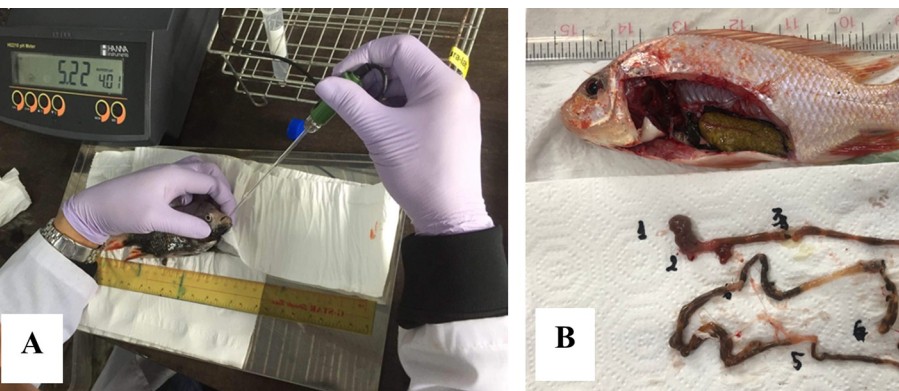

**Fig 1.** pH measurement of gastrointestinal tract of tilapia: A–Mouth; B– 1. Anterior stomach, 2. Posterior stomach; 3,4,5,6: intestine.

The pH of the gastro-intestinal tract (GIT) contents of tilapia was measured using a HI 2210–02 portable pH meter (Hanna, Taiwan) and a 3-mm diameter pH probe (H1095B micro-electrode, Hanna Instruments, Scientific Laboratory Supplier, UK). Tilapia were euthanized before or after feeding (1.5 h later) by exposure to an overdose of Aqui-S in an immersion bath (Bayer, Vietnam) at a concentration of 150 mg L$^{-1}$ for 60 minutes as per [39]. Within the next 5–10 min, the pH of the oral cavity (anterior and posterior mouth) was measured for 5 fish per group (Fig 1A). Then, the GIT was removed and cut into sections to enable measurement of pH at multiple points along its length, namely anterior stomach, posterior stomach, and at four points in the intestine (Fig 1B).

To evaluate the efficacy of GBS encapsulated in Eudragit E100 microparticles as oral vaccine, three treatment groups of a 100 fish each were used, i.e. a negative control group that received mock vaccination with PBS; a delivery system control group that received 0.2 mL of a suspension of blank Eudragit microparticles per fish; and a vaccine group that received 0.2 mL of a suspension of GBS-loaded Eudragit microparticles per fish. The number of fish per treatment group was calculated based on the study of Shoemaker *et al.* [40] and using an on-line calculator [41]. To maintain appropriate stocking density, fish were housed in groups of 25 per tank, with 4 tanks per treatment group. To limit the number of fish used, in line with the 3R principles and because our aim was **not** to see whether GBS encapsulation enhances vaccine efficacy, but rather to test the potential of encapsulated GBS to generate protective efficacy, we did not include vaccination with unencapsulated GBS as a comparator.

Vaccines and control products were administered in the fasted state i.e., one hour prior to the first meal of the day. Before oral administration of the GBS vaccine formulation, the fish were anaesthetized with Aqui-S$^{TM}$ in an immersion bath for 10 minutes. Fish were then orally dosed (0.2mL, by gavage) with PBS, blank or GBS-loaded microparticles (dose of GBS was 5 x 10$^6$ CFU) respectively, using a 1 mL syringe without a needle. All the fish were placed into flow-through experimental tanks (120 L) and observed twice daily for 21 days.

Challenge was conducted by immersion on day 21 post-vaccination with a challenge isolate that had been passaged through naïve tilapia twice to restore pathogen virulence after storage, as previously described in [6]. Fish were transferred to 10 L tanks containing GBS bacteria at $3.5 \times 10^6$ CFU mL$^{-1}$, removed after 30 minutes and placed back into the original flow-through experimental tanks and observed for a further 21 days. The bacterial concentration used for challenge was determined from pilot studies and was designed to give a 70% mortality in the control groups [6].

Throughout the experiment, the fish were fed a commercial tilapia diet (Aquaxcel 7444, Cargill, Vietnam) at 2% of body weight and examined twice daily for gross clinical signs of disease. If moribund or freshly dead fish were found in the tanks, they were killed using a Schedule 1 procedure [42]. On Day 42 of the study, equating to 21 days after challenge, the surviving fish were euthanized using the same Schedule 1 method. Bacterial recovery was attempted from moribund fish and from 3 surviving fish per tank by culturing the head kidney on trypticase soy agar [6].

### 2.4 Data analysis

Data analysis was conducted using Excel and SPSS v 26. Relative percent of survival (RPS) was calculated according to the formula proposed by Amend (1981) [43].

$$RPS = 1 - \left( {cum.mort_{vaccinated}} \big/ {cum.mort_{control}} \right) \qquad (1)$$

Repeated measures ANOVA, followed by post-hoc Tukey was conducted to establish significant differences, if any, in the pH of the gastro-intestinal tract in the fed and fasted states. One way ANOVA, followed by Duncan Test was conducted to compare fish mortality in the different groups after challenge, with $P < 0.05$ deemed to indicate statistical significance.

## 3. Results

### 3.1 Characterization of microparticles

Light and scanning electron microscopy of formalin-inactivated GBS bacteria used as the vaccine showed the characteristic cocci and pair or chain forming features. Chains were generally short, which may be due to formalin exposure and other processing procedures (Fig 2). The

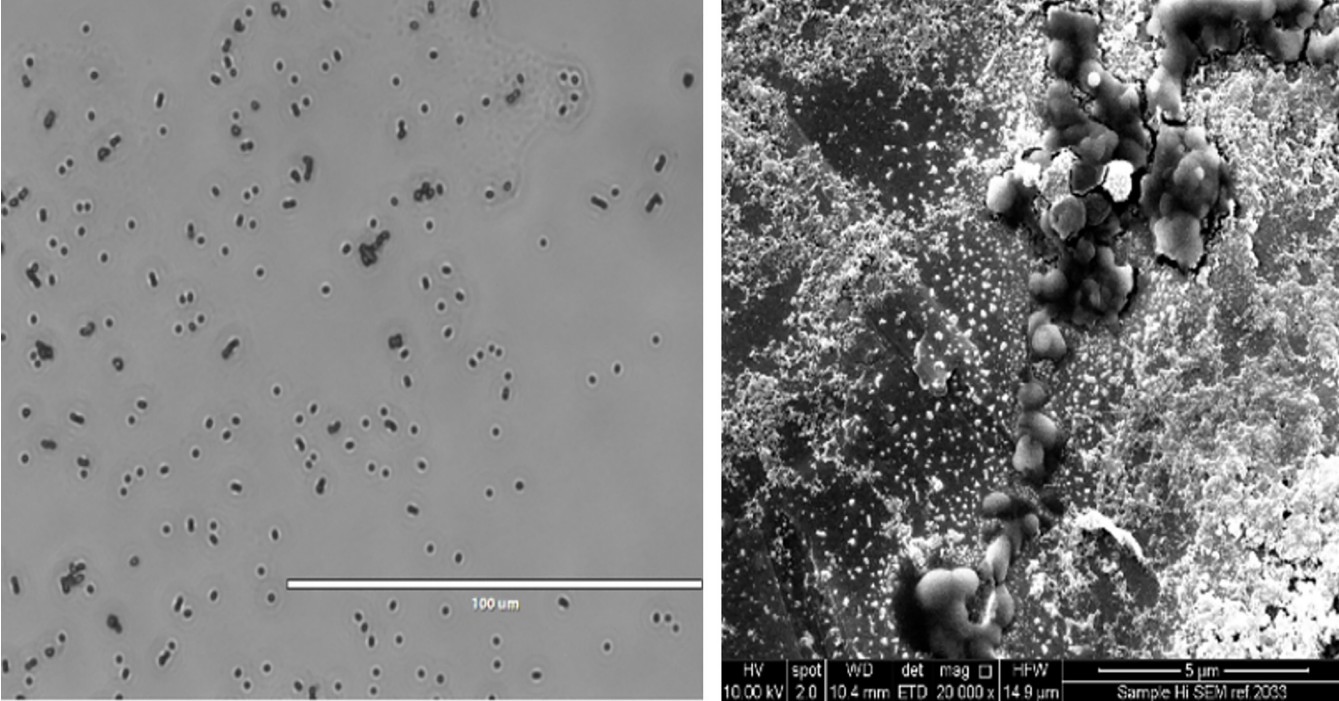

**Fig 2.** Light (A) and scanning electron (Bi-ii) micrographs of the formalin-inactivated bacteria used in vaccine formulations. Scale bar indicates 100 μm (A); 5 μm (B).

**Table 1. Particle size measured by dynamic light scattering (mean ± SD), n = 12.**

|  | Mean particle size (µm) (±SD) | Zeta potential (mV) |
|---|---|---|
| GBS (formalin inactivated) **bacteria** | 2.46 (±0.13) | -23.42 (±1.22) |
| **Blank** Eudragit E100 microparticles | 2.90 (±0.91) | 41.77 (±4.25) |
| **GBS-loaded** Eudragit E100 microparticles | 8.92 (±2.94) | -70.39 (±3.86) |

surface of GBS was negatively charged (Table 1), which is characteristic of bacteria suspended in aqueous media where the various functional groups present on the external cell wall surface will ionize depending on the nature of the external environment [44,45].

Blank Eudragit E100 particles were smooth and spherical and positively charged (Fig 3). In contrast, GBS-loaded particles were negatively charged and were larger with irregular shapes with indentations, which gave the particles the appearance of deflated balls (Table 1, Fig 4).

Upon exposure to 0.1M HCl (simulating the acidic environment in the fasted tilapia stomach), the particle size of both blank and GBS-loaded microparticles decreased within 10 minutes, (Fig 5), indicating rapid dissolution of the polymeric microparticles. Blank Eudragit E100 microparticles dissolved completely, whereas acidic exposure of the GBS-loaded microparticles resulted in reduction in size until a plateau of around 2 µm was reached, indicating release of the entrapped cargo of inactivated GBS.

## 3.2 Tilapia studies

In fasted and fed tilapia, the pH of the GIT contents changed significantly along the GIT from neutral in the mouth, to acidic in the stomach, and close to neutral in the intestines (Fig 6). In

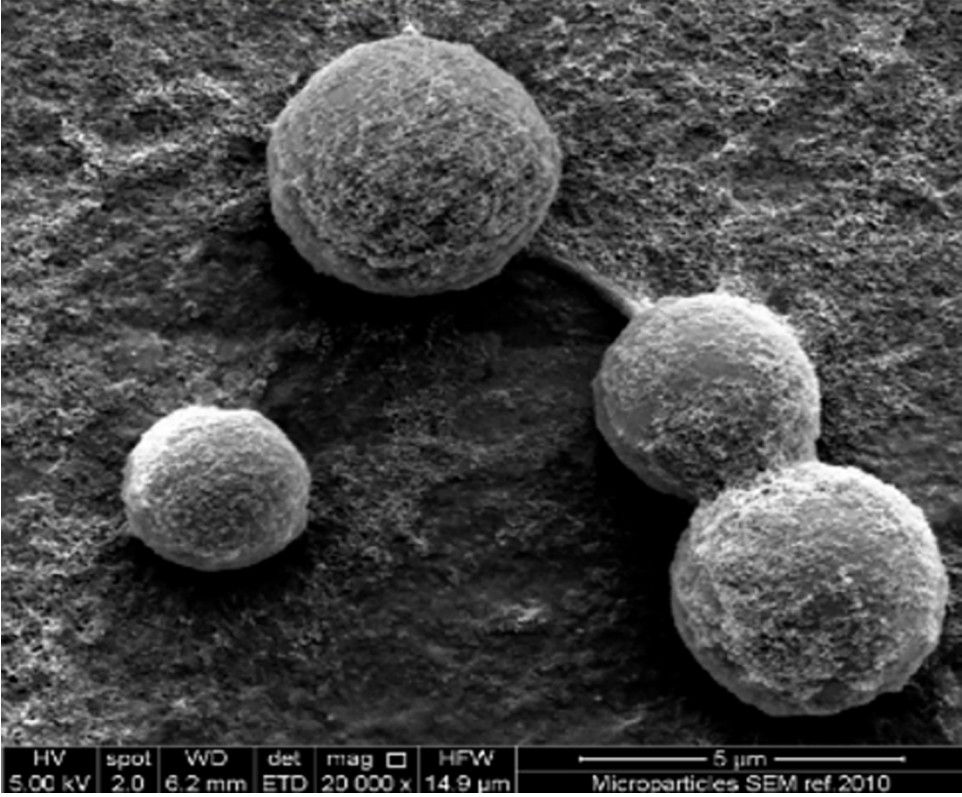

**Fig 3. Scanning electron micrograph of blank Eudragit E100 microparticles.** Scale bar indicates 5 µm.

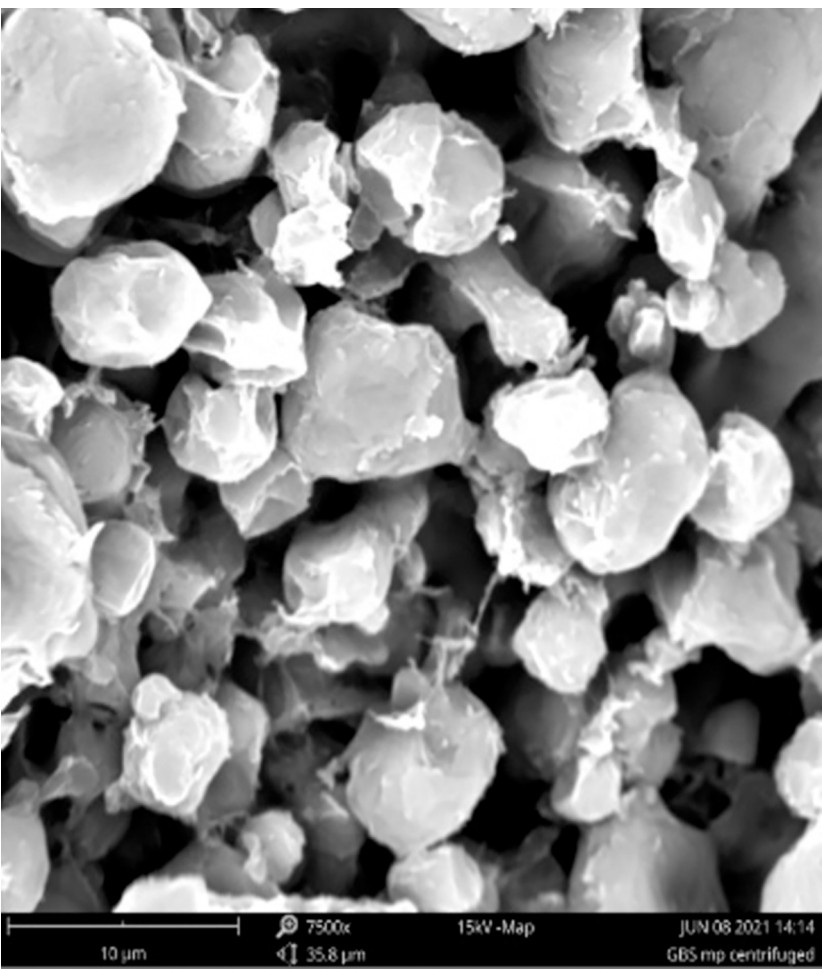

**Fig 4. Scanning electron micrograph of GBS-loaded Eudragit E100 microparticles after freeze-drying.** Scale bar indicates 10 μm.

both fasted and fed states, the stomach pH was significantly lower than the pH in the other sections of the GIT (repeated measures ANOVA with post-hoc comparison, P < 0.05), providing the only environment where Eudragit E100 would dissolve.

The challenge study was successful, as demonstrated by typical signs of *S. agalactiae* infection, including unilateral or bilateral opacification of the eye and eye haemorrhage (Fig 7), and 70.7 ± 4.2% mortality in the control group (Fig 8).

The group which received the blank Eudragit E100 microparticles had similarly high mortality (67.9 ± 4.6%) (ANOVA, Duncan Test, p>0.05), indicating that blank microparticles do not induce protection against *S. agalactiae* challenge. In contrast, mortality in the vaccinated group was only 22.6 ± 4.6% (ANOVA, Duncan Test, p<0.05), equating to RPS = 0.70, and indicating highly successful protection against homologous challenge. The difference among the groups was also noticeable in the onset of mortality, which started at day 2 post challenge in both controls but not until day 4 post-challenge in the vaccine group, and in the last observed mortality, which occurred on day 12 for the vaccinated group but continued to day 14 in both control groups. Pure cultures of bacteria identified as *S. agalactiae* were recovered from 100% of moribund and fresh dead fish. No clinical signs of disease or bacteria were observed or recovered from any of the survivors.

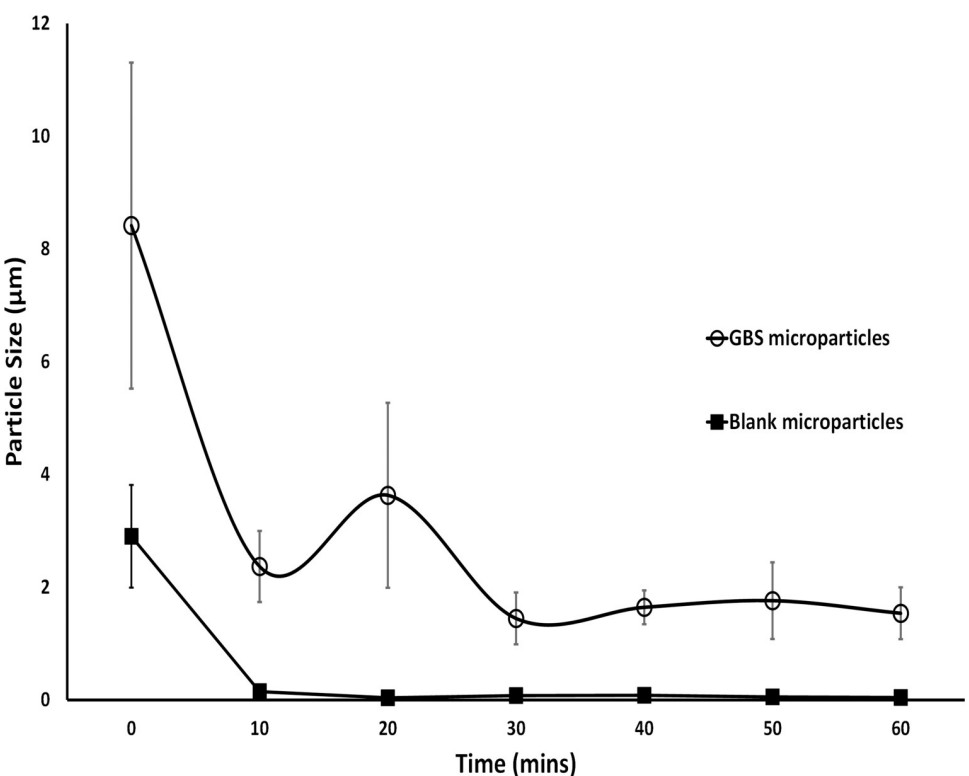

**Fig 5. Change in particle size upon exposure to 0.1M HCl, which is simulating the acidic environment in a fasted tilapia stomach.** GBS-loaded Eudragit E100 microparticles (○); Blank Eudragit E100 microparticles (■); Means are shown. Error bars reflect the SD. N = 12.

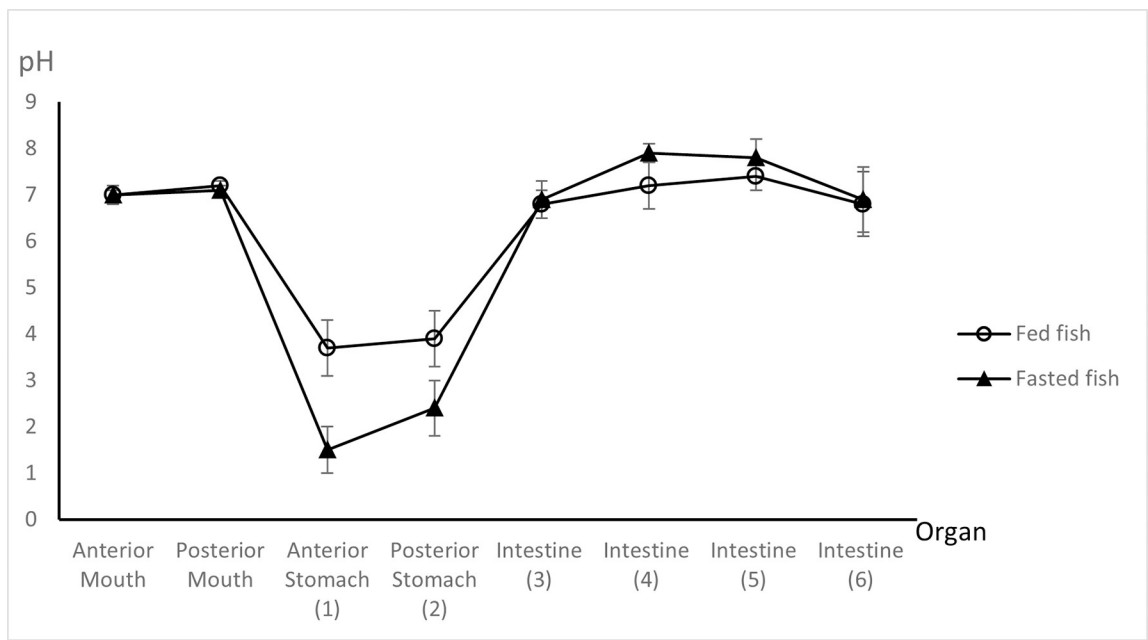

**Fig 6. pH along the gastrointestinal tract of fasted and fed tilapia (mean ± SD), N = 5.**

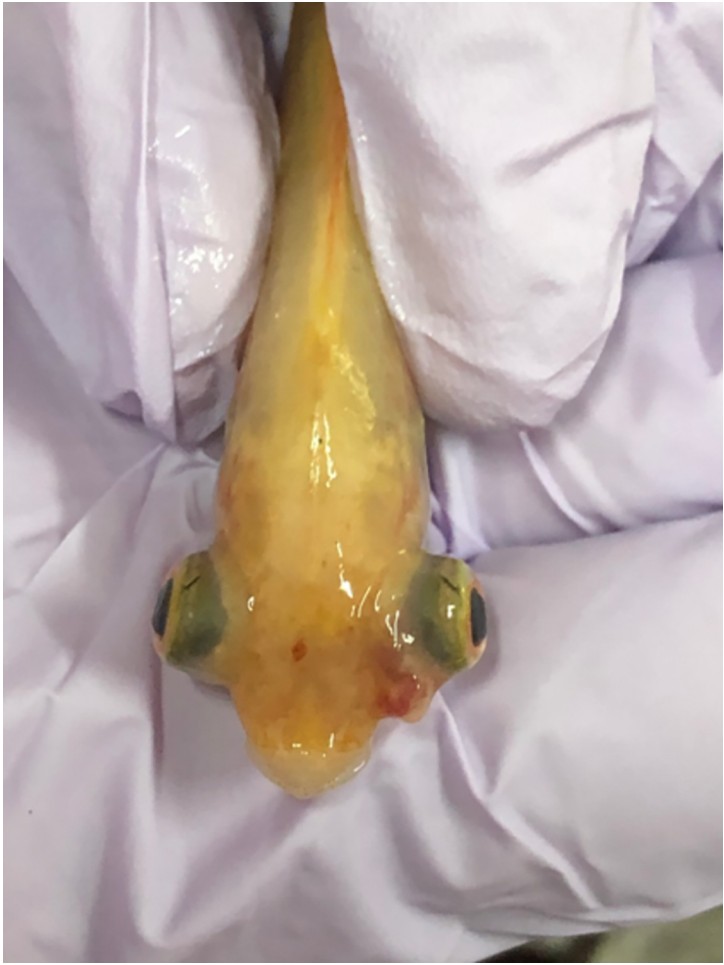

**Fig 7. Bilateral exophthalmia of the eye (pop-eyes) and eye hemorrhages (left eye) were found in moribund fish with *Streptococcus agalactiae* infection.**

## 4. Discussion

This study shows that pH-responsive Eudragit E100 is a promising polymer for preparing a GBS vaccine formulation that enables targeted delivery to the GIT of tilapia and induces a protective immune response. Eudragit E100 performed as expected, i.e. microparticles composed of this polymer dissolved in an acidic environment and released their vaccine cargo, which gave rise to protective immunity. The generally recognized as safe status of the polymer means that adoption of our vaccine platform by the fish farming industry would not meet regulatory hurdles regarding the polymer.

Although we did not measure antibody responses to the vaccine, we clearly demonstrated that the vaccine induced protective immunity, with an RPS of 70%. This compares favourably with other studies. Embregts and Forlenza [32], reviewed oral GBS vaccination studies in tilapia and report 25% to 85% protection, whereby the higher values were generally achieved with multiple vaccine doses administered, and only one vaccine regime in one study achieved 100% protection [32]. Protection of 70% or more, generally required more vaccine administrations or a higher vaccine dose (CFU/fish) than those used in our protocol. Here, we achieved an

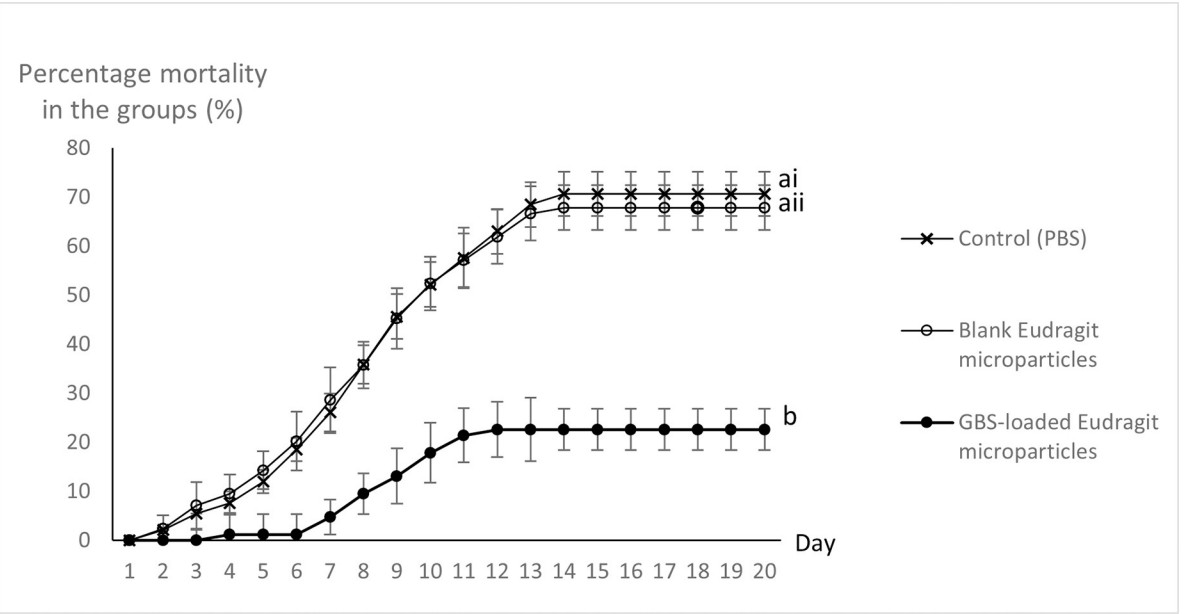

**Fig 8.** Cumulative percentage mortality in the 3 groups following oral gavage with: (ai) PBS in the negative control group; (aii) Blank Eudragit E100 microparticles; (b) GBS-loaded Eudragit E100 microparticles, and challenge by immersion using a homologous strain of Streptococus agalatiae sequence type 283 on day 21 post-gavage. Error bars represent SD from 4 replicate tanks (25 fish per tank) for each of 3 treatment groups.

RPS of 70% after a single oral vaccine administration of only $10^6$ CFU/fish. Most evaluations of vaccine efficacy use intraperitoneal challenge models, which do not reflect the natural route of exposure nor immuno-protective mechanisms that occur at mucosal host-pathogen interfaces. The high RPS observed in our study after a single vaccine dose may be due, at least in part, to the fact that both vaccination and challenge targeted mucosal portals of entry.

Teleost fish have a morphologically and functionally different gut associated lymphoid system (GALT) than mammals, with a distinct lack of a concentrated lymphoid tissue mass such as Peyer's patches found in humans. Instead, lymphoid tissue in the fish gut is diffuse throughout the GIT, consisting mainly of isolated lymphoid patches. In general, development of efficacious oral vaccine is considered challenging due to antigen breakdown in the harsh gastric environment and the potential of antigenic tolerance developing in the gut [32], as well as a general lack of understanding of how tilapia immune system functions [46,47]. Our study based on encapsulation technology sheds some light on how to overcome some of the challenges and bottlenecks of oral vaccine development for fish.

While oral administration of GBS-loaded Eudragit E100 microparticles was effective at reducing mortality upon pathogen challenge, oral administration of blank microparticles resulted in similar mortality compared to fish which only received the buffer (control group, Fig 7). This shows that microparticles by themselves were not responsible for any protection.

We have shown that protective immunity against a homologous challenge strain is possible when formalin-inactivated GBS is liberated from the Eudragit E100 polymeric carrier and released into the acidic environment of the fish stomach. To develop the findings from this investigation, further studies must be conducted to determine whether encapsulated vaccine is more efficacious than unencapsulated vaccine, and to evaluate the efficacy of the encapsulated GBS when incorporated in fish feed. The latter is crucial because feed-based vaccine is the most practical, economical and welfare friendly method of administering vaccines to tilapia.

Other avenues for potential refinement would be the inclusion of an adjuvant to stimulate mucosal immunity and simplification of the GBS-inactivation protocol. If heat-killed GBS could be used rather than formalin-fixed GBS, this would simplify vaccine production and reduce the cost of goods, which is an important consideration for production of economically viable vaccines. There are three major lineages of GBS that can cause streptococcosis in tilapia [48,49], and the ideal aquatic GBS vaccine should provide protection from all three. Commercial vaccines are currently mostly sold as providing protection against biotype 1 (non-haemolytic; equivalent to serotype Ib, clonal complex (CC) 552 [48]) or biotype 2 (haemolytic; serotype Ia, CC7 and serotype III, CC283 [48]) with limited cross-protection between biotypes. In the current study, we evaluated protection against homologous challenge (ST283 used as vaccine strain and as challenge strain). Further work is needed to evaluate whether mono- or multi-valent vaccines administered orally using pH-responsive particles could provide protection against heterologous challenge or multiple strains. Finally, our challenge model was based on immersion, which is a more natural route of exposure than intraperitoneal injection, but not as natural as cohabitation with infected fish. Thus, evaluation of vaccine efficacy in a cohabitation challenge model could be considered as an intermediate step between evaluation based on immersion exposure and field studies.

Although our proof-of-concept study is based on red tilapia, the acidic pH in the stomach and neutral-to-weakly alkaline pH in the intestine is similar to that of other tilapia species (e.g. *Tilapia guineensis*, *Sarotherodon melanothron*) and grey mullet (*Liza falcipinnis*) [50], suggesting that similar pH-sensitive particles may be useful for oral vaccination in those species.

## 5. Conclusions

In this manuscript, we describe the development, and show proof-of-concept of an oral vaccine for the prevention of streptococcosis in tilapia. The vaccine formulation, composed of pH-responsive microparticles entrapping killed *S. agalactiae*, provided very good immune protection against homologous immersion challenge in tilapia. Further vaccine refinements, notably in-feed administration, and evaluation of costs and benefits of the pH-responsive particles are logical next steps in the development of this concept, which may also have applicability across other pathogen- and host- species.

## Author Contributions

**Conceptualization:** Nguyen Ngoc Phuoc, Ruth N. Zadoks, Sudaxshina Murdan.

**Data curation:** Sudaxshina Murdan.

**Formal analysis:** Shazia Bashir, Nguyen Ngoc Phuoc, Ruth N. Zadoks, Sudaxshina Murdan.

**Funding acquisition:** Nguyen Ngoc Phuoc, Abdul Basit, Ruth N. Zadoks, Sudaxshina Murdan.

**Investigation:** Shazia Bashir, Nguyen Ngoc Phuoc.

**Methodology:** Shazia Bashir, Nguyen Ngoc Phuoc, Abdul Basit.

**Project administration:** Nguyen Ngoc Phuoc, Sudaxshina Murdan.

**Supervision:** Nguyen Ngoc Phuoc, Ruth N. Zadoks, Sudaxshina Murdan.

**Writing – original draft:** Shazia Bashir, Nguyen Ngoc Phuoc.

**Writing – review & editing:** Tharangani Herath, Ruth N. Zadoks, Sudaxshina Murdan.

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
