## [Decision Letter · Decision Letter 0]

26 Sep 2022

PONE-D-22-20929An oral pH-responsive Streptococcus agalactiae vaccine formulation provides protective immunity to pathogen challenge in tilapia: a proof-of-concept studyPLOS ONE

Dear Dr. Murdan,

Thank you for submitting your manuscript to PLOS ONE. After careful consideration, we feel that it has merit but does not fully meet PLOS ONE’s publication criteria as it currently stands. Therefore, we invite you to submit a revised version of the manuscript that addresses the points raised during the review process.

We look forward to receiving your revised manuscript.

Kind regards,

Mahmoud Abdel Aziz Mabrok, PhD

Academic Editor

PLOS ONE

Journal Requirements:

 "SM received an internal UCL award funded through Research England’s ‘QR Global Challenges Research Fund’ (https://www.ucl.ac.uk/) for staff (SB) and experiments conducted at UCL and at Hue University.  Furthermore,  UCL,  Hue University (https://hueuni.edu.vn/portal/en/), The University of Sydney (https://www.sydney.edu.au/) and Harper Adams University (https://www.harper-adams.ac.uk/) further funded SM’s, NNP’s, RZ’s and TH’s time on the project. There were no grant numbers. The funders had no role in study design, data collection and analysis, decision to publish, or preparation of the manuscript."

Reviewers' comments:

Reviewer's Responses to Questions

**Comments to the Author**

1. Is the manuscript technically sound, and do the data support the conclusions?

Reviewer #1: Yes

Reviewer #2: Yes

Reviewer #3: Yes

2. Has the statistical analysis been performed appropriately and rigorously? 

Reviewer #1: Yes

Reviewer #2: Yes

Reviewer #3: Yes

3. Have the authors made all data underlying the findings in their manuscript fully available?

Reviewer #1: Yes

Reviewer #2: Yes

Reviewer #3: Yes

4. Is the manuscript presented in an intelligible fashion and written in standard English?

Reviewer #1: Yes

Reviewer #2: Yes

Reviewer #3: Yes

5. Review Comments to the Author

Reviewer #1: This is an interesting study detailing a pH-responsive carrier with formalin-killed whole cell oral vaccine against streptococcus agalactiae ST283 infection in tilapia. However, there are several concerns listed below, for the authors to improve the manuscript.

Major concerns:

1. The Introduction and Discussion sections should be rewritten because authors confused these two parts. Introduction should be included the context, the scope and the importance of your research. In the Discussion chapter, you need summarized the key findings, did not repeat all the results and related the importance of these key findings with the existing scholarly works and so on.

2. The authors should try to acknowledge the limitations of the study in the Discussion chapter, especially in antibody detection and immune response after vaccination.

3. All the figures resolution was not high.

Minor comments:

1. In line 50, Keywords should be “oral vaccine, tilapia, streptococcus agalactiae, microparticles”

2. In line 251, author said “Mean body weight of the tilapia used in this study was 15 ± 2 g”, Why select this age for vaccination? Add references.

3. In line 517-8, “further studies must be conducted to evaluate the efficacy of the encapsulated GBS when incorporated in fish feed and compare to unencapsulated GBS”. Capsular is widely accepted as important protective antigen of Streptococci, so this sentence should be deleted.

Reviewer #2: Authors should revise the manuscript as in the attached word file.

Line 178: Authors should add reference or NCBI GenBank accession number.

Please revise italics in the reference section

All figures should be of high resolution

Reviewer #3: Dear Editor

Greeting,

The manuscript under title ( An oral pH-responsive Streptococcus agalactiaevaccine formulation provides protective immunity to pathogen challenge in tilapia: a proof-of-concept study ).

This study provides good findings that are helpful in minimizing the use of

antimicrobials and antibiotic resistance in aquaculture using oral vaccine . However, I found some points should be clarified.

Constructive Comments to the author

-The language of the text needs more revisions.

Specific comments

Abstract

The abstract section should be briefly introduce the research background and research significance and clarify the research methods, then introduce the main research results, and finally, give the corresponding conclusions. I hope to make a comprehensive modification.

Line 27-30 Add this paragraph to the introduction

Introduction

The novelty should be highlighted more in the introduction. Additionally , organize it to be simple.

Materials and methods

Organize the materials to be simple in writing.

Line 169 Improve this title

Line 176 Put title for this paragraph

Line 191 what do you mean 0.7ml at 106 cfu/mL ? especially at . Did you mean from? How do you adjust it?

Line 244 you wrote, Red tilapia (Oreochromis sp.) was purchased. What is the name of the place ?

Results

The authors can improve the results by rewriting some sentences. Please, highlight just the main results to keep it short.

Discussion

The discussion should concatenate the results of the entire experiment into a reasonable story, but the discussion here is too scattered and not focused.

Add updated references for the paragraphs

Line 439 - 453 Add references to this paragraph. However, it is a long paragraph .

6. PLOS authors have the option to publish the peer review history of their article (what does this mean?). If published, this will include your full peer review and any attached files.

Reviewer #1: No

Reviewer #2: No

Reviewer #3: **Yes: **Hala Fouad Ayoub

---

## [Author Response · Author response to Decision Letter 0]

3 Nov 2022

We thank the reviewers for their constructive comments, and we have revised the manuscript extensively. Changes are shown in red font where possible. Much text has been removed so, this change cannot be shown. Also, to reduce the amount of red font, only the beginning of certain paragraphs that were changed are shown in red. 

We have addressed all PLOS ONE's style requirements, apart from one. We have used Vancouver for the reference style as per PLOS ONE requirements. However, as we are using Endnote Web, the citations are coming as (X), rather than [X] i.e. square brackets. Endnote Web does not have the capability to change () to []. We could use ‘Numbered’ endnote style, but then we don’t get the first 6 authors listed in the bibliography. So we had to keep to Vancouver style. Would it be possible for the desk editors to convert the () into []? We did not do this ourselves as this would entail converting citations to plain text, in which case, we would lose the reference links.

 "SM received an internal UCL award funded through Research England’s ‘QR Global Challenges Research Fund’ (https://www.ucl.ac.uk/) for staff (SB) and experiments conducted at UCL and at Hue University. Furthermore, UCL, Hue University (https://hueuni.edu.vn/portal/en/), The University of Sydney (https://www.sydney.edu.au/) and Harper Adams University (https://www.harper-adams.ac.uk/) further funded SM’s, NNP’s, RZ’s and TH’s time on the project. There were no grant numbers. The funders had no role in study design, data collection and analysis, decision to publish, or preparation of the manuscript."

This is now implemented in the cover letter.

data is now available at

Murdan, Sudax; Phuoc, Nguyen Ngoc; Basit, Abdul; N. Zadoks, Ruth; Herath, Tharangani; Bashir, Shazia (2022): An oral pH-responsive Streptococcus agalactiae vaccine formulation provides protective immunity to pathogen challenge in tilapia. University College London. Dataset. https://doi.org/10.5522/04/19369376.v1

Also stated in the revised cover letter.

The name of the ethics committee who approved the study is added in the main body. Written consent was also obtained from University College London, regarding the conduct of experiments in fish.

Reviewers' comments:

Reviewer #1: This is an interesting study detailing a pH-responsive carrier with formalin-killed whole cell oral vaccine against streptococcus agalactiae ST283 infection in tilapia. However, there are several concerns listed below, for the authors to improve the manuscript.

Major concerns:

1. The Introduction and Discussion sections should be rewritten because authors confused these two parts. Introduction should be included the context, the scope and the importance of your research. In the Discussion chapter, you need summarized the key findings, did not repeat all the results and related the importance of these key findings with the existing scholarly works and so on.

The introduction and discussion have been rewritten and refocused in accordance with the feedback from Reviewers 1 and 3.

2. The authors should try to acknowledge the limitations of the study in the Discussion chapter, especially in antibody detection and immune response after vaccination.

We have added to the discussion chapter that antibody profiles were not determined. It is not known whether vaccine-induced immunity is purely anti-body mediated, so we considered a protective immune response, as observed in our study, a more important read-out.

3. All the figures resolution was not high.

This is now addressed.

Minor comments:

1. In line 50, Keywords should be “oral vaccine, tilapia, streptococcus agalactiae, microparticles”

Implemented as suggested

2. In line 251, author said “Mean body weight of the tilapia used in this study was 15 ± 2 g”, Why select this age for vaccination? Add references.

This is addressed. Reference has been added.

3. In line 517-8, “further studies must be conducted to evaluate the efficacy of the encapsulated GBS when incorporated in fish feed and compare to unencapsulated GBS”. Capsular is widely accepted as important protective antigen of Streptococci, so this sentence should be deleted.

There appears to be confusion between encapsulation of a vaccine to make it pH responsive, and the capsular antigens on the surface of the bacteria. We have rephrased the text to avoid such confusion. Interestingly, capsule is an important virulence factor in human GBS but not in animal GBS. We note this here for the reviewer’s interest but have not added this to the manuscript because that addresses a different type of encapsulation. We have used the word ‘entrapment’ to avoid any confusion.

Reviewer #2: Authors should revise the manuscript as in the attached word file.

Thank you for your detailed feedback on our manuscript. To accommodate the remaining reviewers’ suggestions, we rewrote some sections of the text. Where still relevant, we have incorporated your corrections.

Line 178: Authors should add reference or NCBI GenBank accession number.

An NCBI Genbank accession number was not provided because this isolate has not been sequenced. Use of field isolates to develop challenge models is customary in peer-reviewed tilapia research.

Please revise italics in the reference section

Implemented as suggested.

All figures should be of high resolution

This is now addressed.

Reviewer #3: Dear Editor

Greeting,

The manuscript under title ( An oral pH-responsive Streptococcus agalactiaevaccine formulation provides protective immunity to pathogen challenge in tilapia: a proof-of-concept study ).

This study provides good findings that are helpful in minimizing the use of

antimicrobials and antibiotic resistance in aquaculture using oral vaccine . However, I found some points should be clarified.

Thank you for your positive feedback and your suggestion to improve accessibility of the information. 

Constructive Comments to the author

-The language of the text needs more revisions.

Revisions have been made to all sections of the text, including Abstract, Introduction, Methods, Results and Discussion.

Specific comments

Abstract

The abstract section should be briefly introduce the research background and research significance and clarify the research methods, then introduce the main research results, and finally, give the corresponding conclusions. I hope to make a comprehensive modification.

The abstract has been rewritten using the suggested structure.

Line 27-30 Add this paragraph to the introduction

This information is covered in the introduction.

Introduction

The novelty should be highlighted more in the introduction. Additionally , organize it to be simple.

We have simplified the introduction so that it takes the reader through the importance of intensification of aquaculture for food security, and the resultant risk of disease emergence, to the need for disease control measures, which include antimicrobials and vaccines. Both have disadvantages and limitations, which are summarized briefly, leading logically to the need for better options, e.g., targeted oral vaccination. The current state of knowledge around this option is introduced in such a way that the novelty of our approach is highlighted more clearly. The introduction ends with a shortened and simplified description of the study’s aim, design, and key findings. 

Materials and methods

Organize the materials to be simple in writing.

We have simplified the organization and the writing and checked with non-native speakers whether they found the text easy to understand. 

Line 169 Improve this title

This title has been improved

Line 176 Put title for this paragraph

The improved title for the previous paragraph now also covers this paragraph.

Line 191 what do you mean 0.7ml at 106 cfu/mL ? especially at . Did you mean from? How do you adjust it?

No, we mean at a concentration of 106 CFU/mL. 

0.7 mL was written in the text as 0.7 ml of this GBS suspension was placed in a Cuvette, which was then placed in the instrument, and measurements were then taken. We have removed 0.7ml from the text now to avoid any confusion and as it is not needed. Concentration was adjusted by dilution, prior to placement of the suspension in the cuvette.

Line 244 you wrote, Red tilapia (Oreochromis sp.) was purchased. What is the name of the place ?

Name of the place, Provincial Breeding Centre of Thua Thien Hue at Cu Chanh Commune, Thua Thien Hue province, Vietnam, is in the text.

Results

The authors can improve the results by rewriting some sentences. Please, highlight just the main results to keep it short.

The results section has been shortened, and several sentences have been rewritten.

Discussion

The discussion should concatenate the results of the entire experiment into a reasonable story, but the discussion here is too scattered and not focused.

Discussion has been rewritten and refocused in accordance with the reviewers’ feedback.

Add updated references for the paragraphs

Line 439 - 453 Add references to this paragraph. However, it is a long paragraph.

This paragraph has been removed when the manuscript has been simplified and refocused.

---

## [Decision Letter · Decision Letter 1]

14 Nov 2022

An oral pH-responsive Streptococcus agalactiae vaccine formulation provides protective immunity to pathogen challenge in tilapia: a proof-of-concept study

PONE-D-22-20929R1

Dear Dr. Murdan,

We’re pleased to inform you that your manuscript has been judged scientifically suitable for publication and will be formally accepted for publication once it meets all outstanding technical requirements.

Kind regards,

Mahmoud Abdel Aziz Mabrok, PhD

Academic Editor

PLOS ONE

Additional Editor Comments (optional):

Reviewers' comments:

Reviewer's Responses to Questions

**Comments to the Author**

1. If the authors have adequately addressed your comments raised in a previous round of review and you feel that this manuscript is now acceptable for publication, you may indicate that here to bypass the “Comments to the Author” section, enter your conflict of interest statement in the “Confidential to Editor” section, and submit your "Accept" recommendation.

Reviewer #1: All comments have been addressed

Reviewer #2: All comments have been addressed

2. Is the manuscript technically sound, and do the data support the conclusions?

Reviewer #1: Yes

Reviewer #2: Yes

3. Has the statistical analysis been performed appropriately and rigorously? 

Reviewer #1: Yes

Reviewer #2: Yes

4. Have the authors made all data underlying the findings in their manuscript fully available?

Reviewer #1: Yes

Reviewer #2: Yes

5. Is the manuscript presented in an intelligible fashion and written in standard English?

Reviewer #1: Yes

Reviewer #2: Yes

6. Review Comments to the Author

Reviewer #1: I recommend to accept this manuscription because the authors have improved the contents according to our suggestions

Reviewer #2: (No Response)

7. PLOS authors have the option to publish the peer review history of their article (what does this mean?). If published, this will include your full peer review and any attached files.

Reviewer #1: No

Reviewer #2: No

---

## [Editor Report · Acceptance letter]

16 Nov 2022

PONE-D-22-20929R1 

An oral pH-responsive *Streptococcus agalactiae* vaccine formulation provides protective immunity to pathogen challenge in tilapia: a proof-of-concept study 

Dear Dr. Murdan:

I'm pleased to inform you that your manuscript has been deemed suitable for publication in PLOS ONE. Congratulations! Your manuscript is now with our production department. 

Kind regards, 

on behalf of

Dr. Mahmoud Abdel Aziz Mabrok 

Academic Editor

PLOS ONE